# Risk Factors of Incident Lung Cancer in Patients with Non-Cystic Fibrosis Bronchiectasis: A Korean Population-Based Study

**DOI:** 10.3390/cancers14112604

**Published:** 2022-05-25

**Authors:** Youlim Kim, Kyungdo Han, Juhwan Yoo, Hyung Koo Kang, Tai Sun Park, Dong Won Park, Ji Young Hong, Ji-Yong Moon, Sang-Heon Kim, Tae Hyung Kim, Kwang Ha Yoo, Jang Won Sohn, Ho Joo Yoon, Hayoung Choi, Hyun Lee

**Affiliations:** 1Dvision of Pulmonary and Allergy, Department of Internal Medicine, Konkuk University Medical Center, Konkuk University School of Medicine, Seoul 05030, Korea; weilin810707@gmail.com (Y.K.); 20010025@kuh.ac.kr (K.H.Y.); 2Department of Statistics and Actuarial Science, Soongsil University, Seoul 06978, Korea; hkd917@naver.com; 3Department of Biomedicine and Health Science, Catholic University of Korea, Seoul 06591, Korea; dbwnghks7@naver.com; 4Division of Pulmonary and Critical Care Medicine, Department of Internal Medicine, Ilsan Paik Hospital, Inje University College of Medicine, Goyang 10380, Korea; inspirit26@gmail.com; 5Division of Pulmonary Medicine and Allergy, Department of Internal Medicine, Hanyang University College of Medicine, Seoul 04763, Korea; integrin@hanmail.net (T.S.P.); dongwonpark@hanyang.ac.kr (D.W.P.); respiry@gmail.com (J.-Y.M.); sangheonkim@hanyang.ac.kr (S.-H.K.); drterry@hanyang.ac.kr (T.H.K.); jwsohn@hanyang.ac.kr (J.W.S.); hjyoon@hanyang.ac.kr (H.J.Y.); 6Division of Pulmonary, Allergy, and Critical Care Medicine, Department of Internal Medicine, Hallym University Chuncheon Sacred Heart Hospital, Hallym University College of Medicine, Chuncheon 24253, Korea; mdhong@hallym.or.kr; 7Division of Pulmonary, Allergy, and Critical Care Medicine, Department of Internal Medicine, Hallym University Kangnam Sacred Heart Hospital, Hallym University College of Medicine, Seoul 07442, Korea

**Keywords:** bronchiectasis, lung cancer, risk factor

## Abstract

**Simple Summary:**

Our study evaluated the risk factors of incident lung cancer in subjects with newly diagnosed bronchiectasis, using a large nationwide database. In this study, we showed that male sex, overweight, current smoking, living in rural areas, and comorbid COPD were independently associated with a higher risk of incident lung cancer in participants with bronchiectasis, while mild alcohol consumption was negatively associated with lung cancer development in those with bronchiectasis. To our knowledge, this is the first study assessing the risk factors of lung cancer in patients with bronchiectasis, and it would be helpful for clinicians at real practice.

**Abstract:**

Background: Patients with non-cystic fibrosis bronchiectasis have an increased risk of lung cancer, followed by higher mortality in this population. Because the risk factors of lung cancer have not been well identified, this study aimed to investigate the risk factors of lung cancer in individuals with newly diagnosed bronchiectasis. Methods: This cohort study using the Korean National Health Insurance Service database identified 7425 individuals with incident bronchiectasis among those who participated in the health screening exam in 2009. The cohort was followed from baseline to the date of incident: lung cancer, death, or until the end of the study period. We investigated the risk factors of lung cancer in participants with bronchiectasis using the Cox–proportional hazard models. Results: During median 8.3 years of follow-up duration, 1.9% (138/7425) developed lung cancer. In multivariable analyses, significant factors associated with increased risk of incident lung cancer included: males (adjusted hazard ratio [HR] = 3.54, 95% confidence interval [CI] = 2.17–5.79) than females, the overweight (adjusted HR = 1.55, 95% CI = 1.03–2.35) than the normal weight, current smokers (adjusted HR = 3.10, 95% CI = 2.00–4.79) than never smokers, participants living in the rural area (adjusted HR = 2.54, 95% CI = 1.68–3.85) than those living in the metropolitan area. Among comorbidities, chronic obstructive pulmonary disease was associated with an increased risk of lung cancer (adjusted HR = 1.46, 95% CI = 1.01–2.13) in participants with bronchiectasis. In contrast, mild alcohol consumption was associated with reduced risk of lung cancer (adjusted HR = 0.47, 95% CI = 0.29–0.74) in those with bronchiectasis. Conclusion: This Korean population-based study showed that males, current smoking, overweight, living in rural areas, and comorbid chronic obstructive pulmonary disease are associated with increased risk of lung cancer in individuals with bronchiectasis.

## 1. Introduction

The disease burden of non-cystic fibrosis bronchiectasis (hereafter referred to as bronchiectasis) has been increasing with a substantial rate of morbidity and mortality worldwide [1,2,3,4]. It has been suggested that patients with bronchiectasis have an increased risk of lung cancer [5,6,7], and lung cancer is the leading cause of mortality in patients with bronchiectasis [8,9]. As a result, to decrease the mortality-related burden of bronchiectasis, identifying risk factors of lung cancer in patients with bronchiectasis would be important. 

It seems reasonable that general risk factors of lung cancer such as smoking, socioeconomic factors [10], and personal habits, such as physical activity [11], and respiratory comorbidities, such as previous tuberculosis (TB) or chronic obstructive pulmonary disease (COPD), can also be associated with lung cancer development in patients with bronchiectasis. However, there has been little information regarding risk factors of lung cancer in patients with bronchiectasis. Accordingly, clinical studies that comprehensively evaluate these factors are urgently needed to generate preventive strategies to reduce lung cancer-related disease burden in patients with bronchiectasis.

The Korean National Health Insurance Service (NHIS) database has annual health screening exams, including smoking history, socioeconomic and personal habits [12]. Using this large population-based database, this study aimed to evaluate risk factors of lung cancer in individuals with newly diagnosed bronchiectasis. 

## 2. Methods

### 2.1. Study Population

We conducted a population-based cohort study using the Korean NHIS database [13]. South Korea has a single-payer universal health system that covers almost all Korean citizens; the NHIS maintains claims data on all reimbursed inpatient and outpatient visits, procedures, and prescriptions. Additionally, the NHIS database includes data from annual or biennial health screening exams provided free of charge by the Ministry of Health and Welfare. Approximately 72% of all eligible persons undergo screening [12].

This study initially included 4,234,341 individuals aged ≥ 20 years who participated in the health screening exam between 1 January 2009 and 31 December 2009. We excluded participants who were not diagnosed with bronchiectasis (n = 3,793,117), those diagnosed with bronchiectasis more than one year before the enrollment period (n = 60,582), those diagnosed with cystic fibrosis (n = 292), those diagnosed with any type of cancer (n = 58,943), those who died or were diagnosed with lung cancer within one year after enrollment (n = 10,246), and those with missing data on at least one variable (n = 303,736). Thereafter, we finally identified 7425 participants with incident bronchiectasis (the bronchiectasis cohort). The cohort was followed from baseline to the date of incident lung cancer, death, or until the end of the study period (31 December 2018), whichever came first (Figure 1). 

The Institutional Review Board of Konkuk University Medical Center (application No. KUMC 2022-03-031) approved the study and waived the requirement for informed consent because the NHIS database was constructed after anonymization. This study was conducted in accordance with the Council for International Organization and Medical Sciences (CIOMS) guidelines [14].

### 2.2. Definitions of Bronchiectasis and Lung Cancer

Adult bronchiectasis was defined by the following criteria: (1) age ≥ 20 years; (2) with at least one claim under the International Statistical Classification of Diseases and Related Health Problems, 10th revision (ICD-10) code J47; and (3) excluding those with cystic fibrosis (ICD-10 diagnosis code E84) [1]. 

Lung cancer was defined by ICD-10 diagnosis code C33–C34 plus the specific insurance code for lung cancer (V193) during the follow-up period. In Korea, once a person receives a C-code, he/she is registered to the National Cancer Registry and receives special insurance benefits. Thus, the validity of cancer diagnosis is strictly reviewed by the health insurance review and assessment service [15].

### 2.3. Definitions of Covariates

Smoking status was determined by self-administered questionnaires during the health screening exams, and patients were categorized as never-, ex- or current smokers. Body mass index (BMI) was calculated as weight in kilograms divided by height in meters squared. Patients were categorized according to Asian-specific criteria as underweight (<18.5 kg/m^2^), normal weight (≥18.5–<23 kg/m^2^), overweight (≥23–<25 kg/m^2^), or obese (≥25 kg/m^2^) [16]. Income level was dichotomized at the lowest 25%. The data on alcohol consumption, regular physical activity, and residential area were also determined by self-administered questionnaires. Categories for each variable were as follows: none (0 g/day), mild (<30 g/day), and heavy (≥30 g/day) for alcohol consumption; regular (>30 min of moderate physical at least 5 times per week or >20 min of strenuous physical activity at least 3 times per week), and non-regular for physical activity; metropolitan, city, and rural for the residential area [17]. 

Asthma was defined by ICD-10 code J45–J46, and COPD was defined by ICD-10 code J42–J44 except for J43.0 (unilateral emphysema); the comorbid asthma and COPD were assessed during the enrollment period (the year 2009). TB was defined by the following criteria: (1) presence of ICD-10 code A15–A19, U880, or U881 at least twice in six months between 1 January 2002 and 31 December 2008; or (2) findings of inactive TB, active TB, or suspected TB on chest radiography conducted as part of health screening examinations. Other major comorbidities during 2009 were also defined using the ICD-10 codes [18,19] and summarized using the Charlson Comorbidity Index (CCI) [20,21]. 

### 2.4. Main Outcomes

The main study outcome was the risk factors of incident lung cancer in participants with bronchiectasis.

### 2.5. Statistical Analyses

Baseline characteristics were compared using a χ2 test for categorical variables. The incidence rate of lung cancer was calculated by dividing the number of incident cases by the total follow-up duration (1000 person-years). A Kaplan–Meier plot was used to describe the incidence of lung cancer according to the presence and absence of bronchiectasis, and a log-rank test was used to evaluate significant differences between the two groups. 

To evaluate factors associated with an increased risk of lung cancer, Cox proportional hazards models were used. In multivariable analyses, age, sex, BMI, smoking history, alcohol consumption, income, and residential area were adjusted for in Model 1; variables included in Model 1 and CCI were adjusted for in Model 2; and variables included in Model 2, asthma, COPD, and previous TB history were adjusted for in Model 3. Statistical analyses were performed using SAS version 9.4 (SAS Institute Inc., Cary, NC, USA). All tests were two-sided, and *p*-values < 0.05 were considered statistically significant.

## 3. Results

### 3.1. Baseline Characteristics 

After a median duration of follow-up of 8.3 years (interquartile range, 8.1–8.6 years), the overall incidence of lung cancer was 1.9% in the bronchiectasis cohort (n = 138/7425). As shown in Table 1, participants with bronchiectasis who developed lung cancer were more likely to be older, male, ex-/current smoker, and live in the rural area than those who did not develop incident lung cancer (all *p* < 0.001). Participants who developed lung cancer had significantly higher CCI of ≥ 2 than those who did not develop lung cancer (76.0% vs. 60.5%, *p* < 0.001). Although there were no significant intergroup differences in asthma and TB, participants who developed lung cancer had COPD more frequently than their counterpart (35.5% vs. 16.6%; *p* < 0.001).

### 3.2. Risk Factors of Lung Cancer Development

Multivariable analyses revealed that sex, smoking and alcohol habits, BMI, residential areas, and COPD were significantly associated with the risk of lung cancer in participants with bronchiectasis. Males had a significantly higher risk of incident lung cancer than females (unadjusted hazard ratio [HR] in Model 1 = 3.94, 95% confidence interval [CI], 2.62–5.94; adjusted HR in the fully-adjusted Model 3 = 3.54, 95% CI = 2.17–5.79). Compared with participants with normal BMI, the overweight had a 1.55-fold (95% CI = 1.03–2.35) risk of lung cancer in Model 3; however, there was no significant association between other underweight or obese groups and the risk of lung cancer. Although ex-smokers showed an increased risk of incident lung cancer only in Model 1 (unadjusted HR = 2.05, 95% CI = 1.30–3.25) relative to never-smokers, current smokers showed an increased risk of lung cancer relative to never-smokers (unadjusted HR = 3.80, 95% CI = 2.62–5.51; adjusted HR in Model 3 = 3.10, 95% CI = 2.00–4.79). Whereas mild alcohol consumption was associated with reduced risk of lung cancer (unadjusted HR = 0.59, 95% CI = 0.38–0.91; adjusted HR in Model 3 = 0.47, 95% CI = 0.29–0.74), no significant association was observed between heavy alcohol consumption and the occurrence of lung cancer among participants with bronchiectasis.

When it comes to the residential area, participants living in the rural area revealed a significantly higher risk of incident lung cancer than those living in the metropolitan area (unadjusted HR = 3.85, 95% CI = 2.58–5.74; adjusted HR in Model 3 = 2.54, 95% CI = 1.68–3.85). Among comorbidities, COPD was associated with an increased risk of lung cancer (unadjusted HR = 2.94, 95% CI = 2.07–4.16; adjusted HR in Model 3 = 1.46, 95% CI = 1.01–2.13) (Table 2). Figure 2 depicts the cumulative incidence probability of lung cancer development according to smoking history, residential area, and comorbid COPD. 

## 4. Discussion

This study assessed the risk factors of incident lung cancer in participants with newly diagnosed bronchiectasis. To the best of our knowledge, this is the first study evaluating the risk factors of lung cancer in participants with bronchiectasis, using a large nationwide database. We revealed that male sex, overweight, current smoking, living in rural areas, and comorbid COPD were independently associated with a higher risk of incident lung cancer in participants with bronchiectasis, whereas mild alcohol consumption was negatively associated with lung cancer development in those with bronchiectasis.

Chronic inflammation plays a pivotal role in the development of lung cancer in patients with underlying lung diseases (e.g., COPD, asthma, and interstitial lung disease) [22,23,24,25,26]. As an extension of these results, it can be postulated that bronchiectasis—accompanying a chronic inflammation—may also increase the risk of lung cancer [6,27]. In addition to the postulation, as a previous study showed that lung cancer was one of the leading causes of mortality in patients with bronchiectasis [8], developing strategies to reduce lung cancer-related burden in bronchiectasis patients seems clinically relevant. However, unfortunately, there have been few studies assessing factors related to the increased risk of lung cancer in bronchiectasis. From this view, the results of this study would be meaningful to clinicians by providing risk factors of lung cancer in patients with bronchiectasis.

Age is an important risk factor for lung cancer in both general population and bronchiectasis patients [6,28]. Chung et al., showed that a 1-year increase in age is associated with a 5% increase in the risk of lung cancer in patients with bronchiectasis [6]. Similarly, our study showed that elderly participants with bronchiectasis (age ≥ 60 years) have about 3.6-fold increased risk of lung cancer compared with younger subjects with bronchiectasis (age < 60 years) in univariable analysis. Although the statistical significance disappeared in the multivariable analyses, these results should be interpreted with caution since we dichotomize age. Considering that the prevalence of bronchiectasis increases with aging [1], the increased risk of lung cancer in elderly patients with bronchiectasis should be emphasized. Interestingly, our study showed that males have about 3.5-fold increased risk of lung cancer than females. In line with the results, Chung et al., also showed that males have about 2.2-fold increased risk of lung cancer than females [6]. Although the reasons why males with bronchiectasis have an excessive risk of lung cancer are not fully explainable, there might be a few explanations. It is well known that the prevalence of smoking history as well as noxious gas exposure (especially work-related), are substantially higher in males than females in Korea [29]. In addition, the prevalence of TB, a major risk of bronchiectasis, is higher in males than in females in Korea [30]. Accordingly, Korean males with bronchiectasis may have a higher burden of chronic inflammation superimposed by smoking exposure, noxious gas exposure, or TB destruction. This higher burden of chronic inflammation in males with bronchiectasis compared with females might be linked to this sex disparity.

Cigarette smoking is the most important risk factor of lung cancer, and long-term smoking exposure induced the inflammatory cells infiltration and airway epithelial cell deaths [22,31]. Continuous smoking is riskier than never or past smoking in lung cancer development because airway inflammation and carcinogenesis continues in current smokers [28]. Furthermore, a recent publication dealing with the sex differences in lung cancer incidence found that men have a higher incidence of lung cancer than women in the same level of smoking exposure, which denotes that men have a higher susceptibility to lung cancer [32]. In agreement with the previous finding, male sex was independently associated with the increased risk of lung cancer in participants with bronchiectasis, even after adjustment of potential confounding factors, including smoking history. 

Obesity is generally associated with most types of cancer. However, paradoxically, BMI is known to be negatively associated with lung cancer risk [33]. An obesity-associated inflammatory signaling which activates p53 pathway, a well-known anti-tumor pathway, was suggested to play a role [34]. Interestingly, in our study, by contrast to the previous findings, overweight bronchiectasis participants were more likely to develop lung cancer than those with normal weight, whereas obese bronchiectasis subjects did not show an increased lung cancer risk when compared with those with normal weight. However, it is limited to confirm the obesity-lung cancer relationship based on BMI alone [35,36], so future studies are warranted to explore the relationship between obesity, lung cancer, and our study results. 

In this study, participants living in rural areas were more likely to have a higher risk of incident lung cancer than those living in urban areas. Previous studies evaluating the differences in cancer incidence between rural and urban areas also have found that cancer incidence was significantly higher in rural areas than in urban area [37,38]. Although the reasons for this are not clear, the access to healthcare utilization is more limited in rural residents than in urban residents. Thus, there might be a disparity in the effective treatment of bronchiectasis between rural areas and other regions, and long-term inflammatory responses in patients in rural areas may lead to a high risk of lung cancer due to a lack of effective treatment. Another reason for this phenomenon may be low income in rural residents. Due to the lack of information on income in NHIS data, we alternately used type of insurance as an indicator of household income. Thus, there is a possibility that our adjustment for economic status might be insufficient. In addition, as the population of younger residents in rural areas has been decreasing continuously, most of the residents in those areas are older residents [39]. That might be a reason why the incidence of lung cancer was significantly higher in rural areas compared with other regions.

Interestingly, mild alcohol consumption was found to be related to a relatively lower risk of lung cancer in bronchiectasis subjects. In the result of the EAGLE (Environment and Genetics in Lung Cancer Etiology), which studied alcohol consumption and risk of lung cancer, light drinkers who consumed 0.1–4.9 g alcohol per day were associated with a lower risk of lung cancer compared with non-drinkers or heavy drinkers [40]. The study finding was also replicated in another prospective cohort study [41]. Although the mechanism is not fully explainable, one suggested explanation was that mild alcohol consumption has anti-inflammatory and anti-oxidative effects, leading to lower risk of lung cancer [42]. However, as our study may have reverse causality bias due to the choice of nondrinkers as the reference group, the interpretation of this result should be performed cautiously [43].

It was recently revealed that the risk of lung cancer is increased in smoking COPD patients as well as never-smoking COPD patients [44]. Thus, it could be postulated that COPD, itself, could render additional risk of lung cancer development independently of the effect of smoking. In line with this view, our study showed that COPD, even after adjustment of smoking history, could be a risk factor of lung cancer development in patients with bronchiectasis. Unfortunately, due to the lack of data on pulmonary function, laboratory findings, and radiologic findings in NHIS database, we could not evaluate the detailed aspect of the association among COPD, bronchiectasis, and lung cancer. As these data are available in prospective bronchiectasis cohorts [45,46,47,48,49], future studies using these cohorts are warranted to unveil these association.

There are two limitations to this study that should be acknowledged. First, we used ICD-10 codes and a specific insurance code (V193 code) for the diagnosis of lung cancer. The validity of cancer diagnosis is strictly reviewed by the health insurance review and assessment service, and the National Cancer Registry generally provides V193 code when pathologic results are provided. However, in a very few cases in which biopsy or cytology cannot be performed (e.g., seriously ill status or Do-Not-Resuscitate status), the National Cancer Registry provides V193 code after a comprehensive review of medical records. Thus, there might be a chance of misclassification of lung cancer in patients who did not undergo pathological diagnostic approach. Second, we used ICD-10 codes for the diagnosis of bronchiectasis and comorbidities including COPD. Thus, there might be a misclassification, which is a major limitation of all claim-based studies. 

## 5. Conclusions

In conclusion, this Korean cohort study revealed that individuals with bronchiectasis have a higher risk of lung cancer when they are males, current smokers, overweight, living in rural areas, and have a comorbid with COPD. Mild alcohol consumption was associated with a reduced risk of lung cancer in this population.

## Figures and Tables

**Figure 1 cancers-14-02604-f001:**
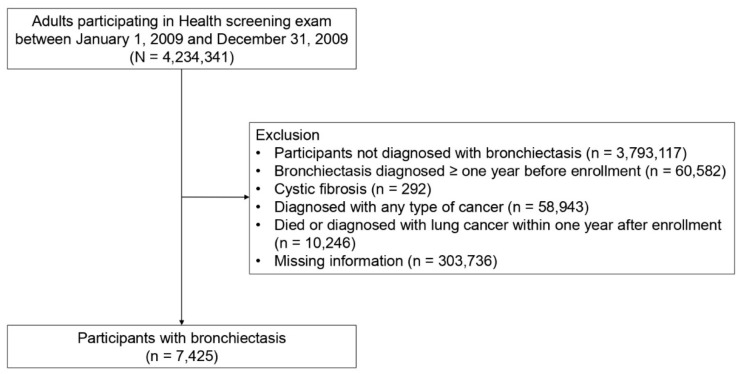
Flow chart of the study population.

**Figure 2 cancers-14-02604-f002:**
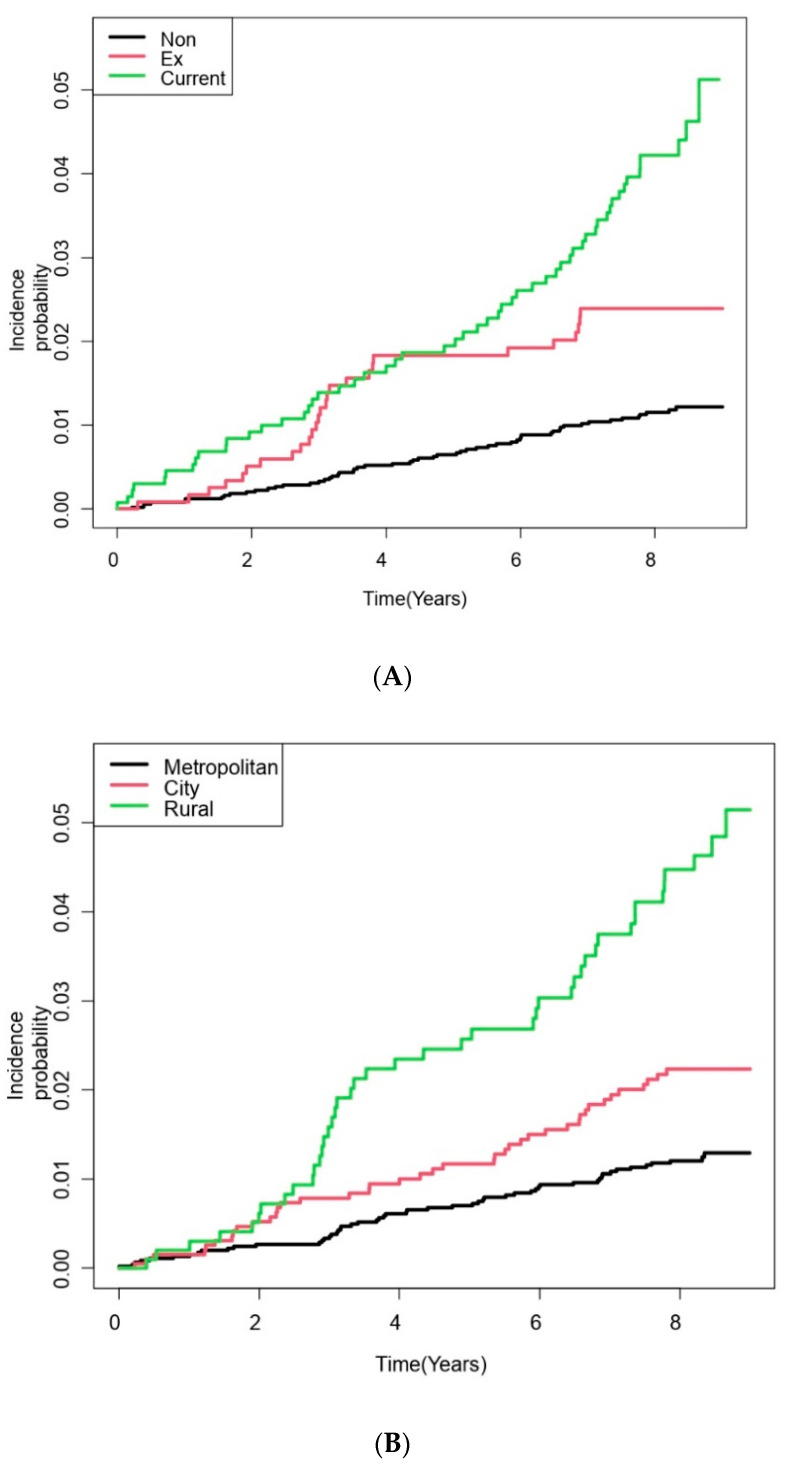
Cumulative incidence probability (/1000 person-years) of lung cancer in participants with bronchiectasis according to risk factors. (**A**) Smoking; (**B**) Residential area; (**C**) Chronic obstructive pulmonary disease.

**Table 1 cancers-14-02604-t001:** Characteristics of the study population.

	Participants with Bronchiectasis (n = 7425)	Incident Lung Cancer		*p*-Value
		No (n = 7286)	Yes (n = 138)	
Age, years				<0.001
<60	3825 (51.5)	3803 (52.2)	22 (15.9)	
≥60	3600 (48.5)	3484 (47.8)	116 (84.1)	
Male sex	3720 (50.1)	3611 (49.6)	109 (79.0)	<0.001
Body mass index				0.215
<18.5	356 (4.8)	348 (4.8)	8 (5.8)	
≥18.5–<23.0	2882 (38.8)	2834 (38.9)	48 (34. 8)	
≥23.0–<25.0	1791 (24.1)	1748 (24.0)	43 (31.2)	
≥25.0	2396 (32.3)	2357 (32.3)	39 (28.2)	
Smoking status				<0.001
Never-smoker	4913 (66.2)	4857 (66.6)	56 (40.6)	
Ever-smoker	1197 (16.1)	1170 (16.1)	27 (19.6)	
Current smoker	1315 (17.7)	1260 (17.3)	55 (39.8)	
Alcohol consumption				0.007
None	4927 (66.4)	4828 (66.3)	99 (71.7)	
Mild	2066 (27.8)	2041 (28.0)	25 (18.1)	
Heavy	432 (5.8)	418 (5.7)	14 (10.2)	
Residential area				<0.001
Metropolitan	4493 (60.5)	4439 (60.9)	54 (39.1)	
City	1940 (26.1)	1899 (26.1)	41 (29.7)	
Rural	992 (13.4)	949 (13.0)	43 (31.2)	
Income				0.200
High	6264 (84.4)	6153 (84.4)	111 (80.4)	
Low	1161 (15.6)	1134 (15.6)	27 (19.6)	
Physical activity				0.701
on-regular	6040 (81.3)	5926 (81.3)	114 (82.6)	
Regular	1385 (18.7)	1361 (18.7)	24 (17.4)	
Charlson Comorbidity Index				0.001
0	571 (7.7)	565 (7.8)	6 (4.4)	
1	2339 (31.5)	2312 (31.7)	27 (19.6)	
≥2	4515 (60.8)	4410 (60.5)	105 (76.0)	
Asthma	2451 (33.0)	2400 (32.9)	51 (37.0)	0.320
COPD	1261 (17.0)	1212 (16.6)	49 (35.5)	<0.001
Tuberculosis	1336 (18.0)	1310 (18.0)	26 (18.8)	0.794

Data are represented as a number (percentage). Definitions of abbreviations: COPD = chronic obstructive pulmonary disease.

**Table 2 cancers-14-02604-t002:** Incidence and hazard ratio of incident lung cancer in participants with bronchiectasis versus those without bronchiectasis with stratified analysis by smoking status.

	Number of Participants	Follow-Up Duration (PY)	Incidence Rate (/1000 PY)	HR (95% CI)			
				Crude Model	Model 1	Model 2	Model 3
Age, years							
<60	3825	31,470	0.699	1 (Reference)	1 (Reference)	1 (Reference)	1 (Reference)
≥60	3600	27,375	4.237	6.11 (3.87–9.64)	1.61 (0.81–3.17)	1.59 (0.81–3.13)	1.61 (0.81–3.18)
Sex							
Female	3705	30,065	0.965	1 (Reference)	1 (Reference)	1 (Reference)	1 (Reference)
Male	3720	28,780	3.787	3.94 (2.62–5.94)	3.60 (2.21–5.86)	3.61 (2.22–5.88)	3.54 (2.17–5.79)
Body mass index							
<18.5	356	2561	3.124	1.48 (0.70–3.14)	1.19 (0.56–2.54)	1.23 (0.58–2.62)	1.21(0.57–2.57)
≥18.5–<23.0	2882	22,674	2.117	1 (Reference)	1 (Reference)	1 (Reference)	1 (Reference)
≥23.0–<25.0	1791	14,329	3.001	1.42 (0.94–2.14)	1.58 (1.05–2.39)	1.58 (1.04–2.39)	1.55 (1.03–2.35)
≥25.0	2396	19,281	2.023	0.95 (0.63–1.46)	1.21 (0.79–1.86)	1.17 (0.76–1.80)	1.21 (0.79–1.86)
Smoking status							
Never smoker	4913	39,361	1.423	1 (Reference)	1 (Reference)	1 (Reference)	1 (Reference)
Ex-smoker	1197	9272	2.912	2.05 (1.30–3.25)	1.25 (0.75–2.10)	1.25 (0.75–2.10)	1.21 (0.72–2.03)
Current smoker	1315	10,213	5.385	3.80 (2.62–5.51)	3.21 (2.08–4.95)	3.22 (2.09–4.97)	3.10 (2.00–4.79)
Alcohol drinking							
None	4927	38,830	2.550	1 (Reference)	1 (Reference)	1 (Reference)	1 (Reference)
Mild	2066	16,639	1.503	0.59 (0.38–0.91)	0.46 (0.29–0.73)	0.46 (0.29–0.73)	0.47 (0.29–0.74)
Heavy	432	3376	4.147	1.63 (0.93–2.85)	0.95 (0.53–1.70)	0.96 (0.54–1.73)	0.94 (0.52–1.69)
Residential area							
Metropolitan	4493	36,001	1.500	1 (Reference)	1 (Reference)	1 (Reference)	1 (Reference)
City	1940	15,336	2.673	1.79 (1.19–2.68)	1.52 (1.01–2.29)	1.50 (0.99–2.25)	1.49 (0.98–2.24)
Rural	992	7508	5.727	3.85 (2.58–5.74)	2.65 (1.76–4.01)	2.64 (1.74–3.98)	2.54 (1.68–3.85)
Income							
High	6264	49,704	2.233	1 (Reference)	1 (Reference)	1 (Reference)	1 (Reference)
Low	1161	9142	2.954	1.32 (0.87–2.02)	1.45 (0.95–2.21)	1.44 (0.95–2.20)	1.44 (0.95–2.21)
Physical activity							
Non-regular	6040	47,718	2.389	1 (Reference)	1 (Reference)	1 (Reference)	1 (Reference)
Regular	1385	11,127	2.157	0.90 (0.58–1.40)	0.94 (0.60–1.47)	0.93 (0.60–1.46)	0.93 (0.60–1.45)
CCI							
0	571	4749	1.263	1 (Reference)		1 (Reference)	
1	2339	18,980	1.423	1.13 (0.47–2.73)		1.06 (0.44–2.56)	
≥2	4515	35,116	2.990	2.38 (1.04–5.41)		1.59 (0.70–3.64)	
Asthma							
No	4974	39,905	2.180	1 (Reference)			1 (Reference)
Yes	2451	18,941	2.693	1.24 (0.88–1.75)			0.91 (0.63–1.31)
COPD							
No	6164	49,514	1.797	1 (Reference)			1 (Reference)
Yes	1261	9331	5.251	2.94 (2.07–4.16)			1.46 (1.01–2.13)
Tuberculosis							
No	6089	48,515	2.309	1 (Reference)			1 (Reference)
Yes	1336	10,331	2.517	1.09 (0.71–1.67)			0.81 (0.52–1.25)

Model 1 is adjusted for age, sex, body mass index, smoking history, alcohol drinking, income, and residential area; Model 2 is adjusted for variables included in the Model 1 and Charlson Comorbidity Index; Model 3 is adjusted for variables included in the Model 2, asthma, COPD, and tuberculosis. Definitions of abbreviations: PY = person-years; HR = hazard ratio; CI = confidence interval.

## Data Availability

The data presented in this study are available on request from the corresponding author.

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
