# Peer review of "Risk Factors of Incident Lung Cancer in Patients with Non-Cystic Fibrosis Bronchiectasis: A Korean Population-Based Study"

_cancers, 2022, doi:10.3390/cancers14112604_

Round 1

Reviewer 1 Report

Kim et al. affirm that co-existence of COPD and bronchiectasis in Korean patients presents higher incidence of lung cancer, thus causing increased mortality in this cohort. The experimental data were presented systematically in a clearly-written manuscript.

Specific comments

Introduction. A brief definition of bronchiectasis and COPD including etiology and its prevalence in Korea is essential. Similarly, the incidence of lung cancer in Korea should also be included.

Results/Discussion. The relevance of age as one of the potential risk factors of lung cancer development may have been overlooked in spite the authors` statement that “participants with (..) were more likely to be older” (see Table 1), a relatively high incidence rate (4.237) compared to <60 years of age) and having a HR of 1.61 in Table 2. Although it may not be statistically significant, this parameter merits discussion as a previous literature affirmed that the incidence of lung cancer in bronchiectasis patients escalates with increasing age in both male and female cohorts. Further, some studies have implicated advance age, in general, as a risk factor for lung cancer.

Discussion/Conclusions. Considering the current data and previous reports (Choi et al 2021, Chung et al 2014), the risk of having lung cancer among the studied populations can be traced back to bronchiectasis, particularly, its coexistence with COPD. It is high time that specific modalities be suggested to ensure prevention and diagnosis of these diseases, and the risk of acquiring lung cancer.

Title. It is highly suggested to include the exact population (Korea) where data of the study were obtained, and as appropriate in the entire manuscript.

Figure 2. Kindly include definitions of LCA and BE.

Author Response

## Response to Reviewer 1

General comment

Kim et al. affirm that co-existence of COPD and bronchiectasis in Korean patients presents higher incidence of lung cancer, thus causing increased mortality in this cohort. The experimental data were presented systematically in a clearly-written manuscript.

Response. We appreciate the reviewer’s words of encouragement and helpful comments. We are submitting a revised manuscript that addresses these concerns. A detailed point-by-point response to these concerns is provided.

Specific comments

Comment 1 (C1). Results/Discussion. The relevance of age as one of the potential risk factors of lung cancer development may have been overlooked in spite the authors` statement that “participants with (..) were more likely to be older” (see Table 1), a relatively high incidence rate (4.237) compared to <60 years of age) and having a HR of 1.61 in Table 2. Although it may not be statistically significant, this parameter merits discussion as a previous literature affirmed that the incidence of lung cancer in bronchiectasis patients escalates with increasing age in both male and female cohorts. Further, some studies have implicated advance age, in general, as a risk factor for lung cancer.

Response 1 (R1). We totally agree with your concern. As recommended, we discussed this issue in the revised manuscript (page 8,3rd paragraph – page 9).

“Age is an important risk factor for lung cancer in both general population and bronchiectasis patients [6,26]. Chung et al., showed that a 1-year increase in age is associated with a 5% increase in the risk of lung cancer in patients with bronchiectasis [6]. Similarly, our study showed that elderly subjects with bronchiectasis (age ≥ 60 years) have about 3.6-fold increased risk of lung cancer compared to younger subjects with bronchiectasis (age < 60 years) in univariable analysis. Although the statistical significance disappeared in the multivariable analyses, these results should be interpreted with caution since we dichotomize age. Considering that the prevalence of bronchiectasis increases with aging [1], the increased risk of lung cancer in elderly patients with bronchiectasis should be emphasized.

C2. Discussion/Conclusions. Considering the current data and previous reports (Choi et al 2021, Chung et al 2014), the risk of having lung cancer among the studied populations can be traced back to bronchiectasis, particularly, its coexistence with COPD. It is high time that specific modalities be suggested to ensure prevention and diagnosis of these diseases, and the risk of acquiring lung cancer.

R2. Thank you for your valuable comment. We have discussed this issue in the Discussion section of the revised manuscript (page 8, 2nd paragraph).

Chronic inflammation plays a pivotal role in the development of lung cancer in patients with underlying lung diseases (e.g., COPD, asthma, and interstitial lung disease) [20-24]. As an extension of these results, it can be postulated that bronchiectasis—accompanying a chronic inflammation—may also increase the risk of lung cancer [6, 25]. In addition to the postulation, as a previous study showed that lung cancer was one of the leading causes of mortality in patients with bronchiectasis [7], developing strategies to reduce lung cancer-related burden in bronchiectasis patients seems clinically relevant.”

C3. Title. It is highly suggested to include the exact population (Korea) where data of the study were obtained, and as appropriate in the entire manuscript.

R3. Thank you for your comment. We have clarified the point in Title, Abstract, and the Discussion section of the revised manuscript (page 1 and page 9).

C4. Figure 2. Kindly include definitions of LCA and BE.

R4. We appreciate the reviewer’s careful review of our manuscript. We have clarified the abbreviations as recommend in the revised manuscript (page 7-8, revised version of figure 2).

Reviewer 2 Report

This manuscript explores the risk factors of incident lung cancer in patients with non-cystic fibrosis bronchiectasis using the National Health Insurance Service database. The authors found that males, current smoking, overweight, living in rural areas, and a comorbid chronic obstructive pulmonary disease are associated with an increased risk of lung cancer in individuals with bronchiectasis. The topic is interesting, and the manuscript is well-organized. However, the following major revisions need to be considered before its consideration for publication.

Specific comments:

  1. In Figure 2, the results should be statistically analyzed to investigate the difference between different groups. In addition, this figure needs to be reformatted to save space.
  2. Make sure to use uniform lettering and sizing in the manuscript. For example, the correspondence email and Background (Abstract) section.
  3. The authors found subjects living in rural areas were more likely to have a higher risk of incident lung cancer than those in urban areas. This is a question worthy of in-depth consideration. Whether the patient received effective treatment in these two areas? There is any difference in the treatment of non-cystic fibrosis bronchiectasis between rural and urban areas? Whether is possible that long-term inflammatory responses lead to a high risk of lung cancer due to a lack of effective treatment in rural areas? In addition, the risk of lung cancer is higher among the elderly (>60 years), are there any differences in the age structure of the population between those two areas? The author can conduct more investigations based on the (electronic) pathological records.
  4. Males have a higher risk of lung cancer compared with females (nearly four times). How to explain this difference? The authors should discuss this in the discussion section.
  5. Authors are recommended to include the information about CIOMS guidelines; whether this epidemiological study adheres to or follows the CIOMS guidelines.
  6. References should be uniform according to the journal format.

Author Response

## Response to Reviewer 2

General comment

This manuscript explores the risk factors of incident lung cancer in patients with non-cystic fibrosis bronchiectasis using the National Health Insurance Service database. The authors found that males, current smoking, overweight, living in rural areas, and a comorbid chronic obstructive pulmonary disease are associated with an increased risk of lung cancer in individuals with bronchiectasis. The topic is interesting, and the manuscript is well-organized. However, the following major revisions need to be considered before its consideration for publication.

Response. We appreciate the reviewer’s words of encouragement and helpful comments. We are submitting a revised manuscript that addresses these concerns. A detailed point-by-point response to these concerns is provided.

Specific comments

C1. In Figure 2, the results should be statistically analyzed to investigate the difference between different groups. In addition, this figure needs to be reformatted to save space.

R1. Thank you for your comment. We have reformatted Figure 2 in the revised manuscript (page 7-8).

C2. Make sure to use uniform lettering and sizing in the manuscript. For example, the correspondence email and Background (Abstract) section.

R2. As recommended, we have modified the issue in the revised manuscript (page 1).

C3. The authors found subjects living in rural areas were more likely to have a higher risk of incident lung cancer than those in urban areas. This is a question worthy of in-depth consideration. Whether the patient received effective treatment in these two areas? There is any difference in the treatment of non-cystic fibrosis bronchiectasis between rural and urban areas? Whether is possible that long-term inflammatory responses lead to a high risk of lung cancer due to a lack of effective treatment in rural areas? In addition, the risk of lung cancer is higher among the elderly (>60 years), are there any differences in the age structure of the population between those two areas? The author can conduct more investigations based on the (electronic) pathological records.

R3. Thank you for pointing this out, which we did not fully acknowledge in our original manuscript. In our original manuscript, we described the phenomenon had been caused by a difference in the access to healthcare utilization between two regions. In addition to this, we agree with the reviewer’s view that the disparity in the effective treatment of bronchiectasis between rural areas and other regions can cause long-term inflammatory responses in bronchiectasis patients in rural area and this can lead to a high risk of lung cancer. We would like to appreciate the reviewer’s insightful comment about age structure. As the population of younger residents in rural areas has been decreasing continuously, most of residents in that area are older residents. That might be another reason why the incidence of lung cancer was significantly higher in the rural area compared with in other regions. We have clarified this point in the Discussion section of the revised manuscript (page 9, 3rd paragraph).

“Thus, there might be a disparity in the effective treatment of bronchiectasis between rural areas and other regions, and long-term inflammatory responses in patients in rural areas may lead to a high risk of lung cancer due to a lack of effective treatment.”

“In addition, as the population of younger residents in rural areas has been decreasing continuously, most of the residents in that area are older residents [37]. That might be a reason why the incidence of lung cancer was significantly higher in rural areas compared with other regions.”

C4. Males have a higher risk of lung cancer compared with females (nearly four times). How to explain this difference? The authors should discuss this in the discussion section.

R4. Thank you for your comment. In the revised manuscript, we discussed this in the Discussion section (page 9, 1st paragraph).

“Interestingly, our study showed that males have about 3.5-fold increased risk of lung cancer than females. In line with the results, Chung et al, also showed that males have about 2.2-fold increased risk of lung cancer than females [6]. Although the reasons why males with bronchiectasis have an excessive risk of lung cancer are not fully explainable, there might be a few explanations. It is well known that the prevalence of smoking history as well as noxious gas exposure (especially work-related), are substantially higher in males than females in Korea [27]. In addition, the prevalence of TB, a major risk of bronchiectasis is higher in males than in females in Korea [28]. Accordingly, Korean males with bronchiectasis may have a higher burden of chronic inflammation superimposed by smoking exposure, noxious gas exposure, or TB destruction. This higher burden of chronic inflammation in males with bronchiectasis compared to females might be linked to this sex disparity.”

C5. Authors are recommended to include the information about CIOMS guidelines; whether this epidemiological study adheres to or follows the CIOMS guidelines.

R5. Thank you for your valuable recommendation. We have clarified that our study had adhered to the Council for International Organizations and Medical Sciences (CIOMS) guidelines in the Methods section of the revised manuscript (page 3, 1st paragraph).

C6. References should be uniform according to the journal format.

R6. As recommended, we have reformatted the References section of the revised manuscript (pages 11–13).

Reviewer 3 Report

Recently I was invited to review an interesting paper entitled: “Risk factors of incident lung cancer in patients with non-cystic fibrosis bronchiectasis” The paper is well written and easy to understand. I appreciate the use of tables and figures which make the text clear. It is important to discuss the new risk factors for lung cancer. Bronchiectases are potential, but not a well-documented risk factor for the development of lung cancer. One must remember that there is a common reason for lung cancer and bronchiectasis (other than related to CF) – cigarette smoking. The authors draw their conclusions on the basis of an analysis of a huge national database. This type of data enables precise and well-planned analysis. However, in this case, the authors did not decide to answer a significant question – is the presence of bronchiectasis a risk factor for NSCLC. There is no control group in the study. Good quality data are wasted and only an observational study was created. I would insist on the authors perform a comparison, preferably with the use of the PSMA method to compare the risk of lung cancer in patients with and without bronchiectasis. If this comparison is not performed, then the study is just a plain presentation of the characteristics of patients with bronchiectasis who develop lung cancer. This is relatively not interesting.

Moreover, I have some additional remarks that should be addressed by the authors.

Major remarks.

What were the criteria for the diagnosis of lung cancer? Was the disease confirmed by a cytology/histology in every case? What criteria were followed to define COPD and bronchiectasis? Is there a specific obligation by standards of care to diagnose those diseases?

Figure 2. Please define the abbreviations used.

Author Response

## Response to Reviewer 3

General comment

Recently I was invited to review an interesting paper entitled: “Risk factors of incident lung cancer in patients with non-cystic fibrosis bronchiectasis” The paper is well written and easy to understand. I appreciate the use of tables and figures which make the text clear. It is important to discuss the new risk factors for lung cancer. Bronchiectases are potential, but not a well-documented risk factor for the development of lung cancer. One must remember that there is a common reason for lung cancer and bronchiectasis (other than related to CF) – cigarette smoking. The authors draw their conclusions on the basis of an analysis of a huge national database. This type of data enables precise and well-planned analysis. However, in this case, the authors did not decide to answer a significant question – is the presence of bronchiectasis a risk factor for NSCLC. There is no control group in the study. Good quality data are wasted and only an observational study was created. I would insist on the authors perform a comparison, preferably with the use of the PSMA method to compare the risk of lung cancer in patients with and without bronchiectasis. If this comparison is not performed, then the study is just a plain presentation of the characteristics of patients with bronchiectasis who develop lung cancer. This is relatively not interesting.

Moreover, I have some additional remarks that should be addressed by the authors.

Response. We sincerely appreciate the reviewer’s comments. We also agree with the reviewer that analyses comparing the risk of lung cancer in individuals with bronchiectasis versus in those without bronchiectasis are very important. However, our team has already performed the analyses (Individuals with bronchiectasis have 22% increased risk of lung cancer compared to those without bronchiectasis even after adjusting for potential confounders), and this work is going to be published in another journal (Annals of American Thoracic Society) soon. We believe that evaluating risk factors related to lung cancer development in individuals with bronchiectasis is also important since this study could provide meaningful data to physicians managing patients with bronchiectasis. In this regard, we would like to suggest that this study has its own value to be published.

Specific comments

C1. What were the criteria for the diagnosis of lung cancer? Was the disease confirmed by a cytology/histology in every case? What criteria were followed to define COPD and bronchiectasis? Is there a specific obligation by standards of care to diagnose those diseases?

R1-1. Lung cancer

Thank you for your comment. Lung cancer was defined by ICD-10 diagnosis code C33–C34 plus the specific insurance code for lung cancer (V193) during the follow-up period. In Korea, once a person receives an ICD-10 code of lung cancer, he/she is registered to the National Cancer Registry and receives special insurance benefits. Thus, the validity of cancer diagnosis is strictly reviewed by the health insurance review and assessment service. Generally, the National Cancer Registry provides V193 code only when pathologic results (cytology or histology) are provided. However, in very few cases in which biopsy or cytology cannot be performed (DNR status or seriously ill status), the National Cancer Registry provides V193 code after a comprehensive review of medical records including medical charts and imaging information (CT, PET-CT, etc.). We added this information in the Discussion section (page10, 2nd paragraph)

“First, we used ICD-10 codes (C33–34) and a specific insurance code (V193 code) for the diagnosis of lung cancer. In Korea, once a person is diagnosed with lung cancer, he or she is registered with the National Cancer Registry and receives special insurance benefits. Thus, the validity of cancer diagnosis is strictly reviewed by the health insurance review and assessment service. Generally, the National Cancer Registry provides V193 code when pathologic results are provided. However, in very few cases in which biopsy or cytology cannot be performed (e.g., seriously ill status or Do-Not-Resuscitate status), the National Cancer Registry provides V193 code after a comprehensive review of medical records. Thus, there might be a chance of misclassification of lung cancer in patients who did not have pathologic results.”

R2-2. Bronchiectasis and COPD

This study defined bronchiectasis and COPD by ICD-10 diagnosis codes. Therefore, the two diseases diagnosis do not require a specific obligation from physicians. Because the operative diagnosis of bronchiectasis and COPD were used in several previous studies, we carefully suggest that the diagnosis of bronchiectasis and COPD might be correct. Nonetheless, we have clarified this misclassification issue as a limitation in the Discussion section of the revised manuscript (page 10, 2nd paragraph).

“Second, we used ICD-10 codes for the diagnosis of bronchiectasis and comorbidities. Thus, there might be a misclassification, which is a major limitation of all-claim based studies”

C2. Figure 2. Please define the abbreviations used.

R2. We have defined the abbreviations used in Figure 2 in the revised manuscript (page 7-8).

Round 2

Reviewer 2 Report

I appreciate that the authors have addressed the comments. The revised version of the manuscript “Risk factors of incident lung cancer in patients with non-cystic fibrosis bronchiectasis: a Korean population-based study” has improved considerably.

Author Response

Comment. I appreciate that the authors have addressed the comments. The revised version of the manuscript “Risk factors of incident lung cancer in patients with non-cystic fibrosis bronchiectasis: a Korean population-based study” has improved considerably.

Response. We appreciate the reviewer for the comprehensive review of our manuscript. We are glad that our revision could meet the standard of the review.

Reviewer 3 Report

I was repeatedly asked to review a paper entitled:  “Risk factors of incident lung cancer in patients with non-cystic fibrosis bronchiectasis”. Unfortunately, I do not find significant improvements in the text after my previous review. I would like to repeat my previous comment that in this case the authors did not decide to answer a significant question – is the presence of bronchiectasis a risk factor for NSCLC. There is no control group in the study. Good quality data are wasted and only an observational study was created. I would insist on the authors to perform a comparison, preferably with the use of PSMA method to compare the risk of lung cancer in patients with and without bronchiectasis. If this comparison is not performed, than the study is just a plain presentation of the characteristics of patients with bronchiectasis who develop lung cancer. This is relatively not interesting. I have some additional remarks that should be addressed by the authors.

Major remarks.

What were the criteria for diagnosis of lung cancer. Was the disease confirmed by a cytology.histology in every case? What criteria were followed to define the COPD and bronchiectasis? Is there a specific obligation by standards of care to diagnose those diseases?

Author Response

Comment 1. I was repeatedly asked to review a paper entitled:  “Risk factors of incident lung cancer in patients with non-cystic fibrosis bronchiectasis”. Unfortunately, I do not find significant improvements in the text after my previous review. I would like to repeat my previous comment that in this case the authors did not decide to answer a significant question – is the presence of bronchiectasis a risk factor for NSCLC. There is no control group in the study. Good quality data are wasted and only an observational study was created. I would insist on the authors to perform a comparison, preferably with the use of PSMA method to compare the risk of lung cancer in patients with and without bronchiectasis. If this comparison is not performed, than the study is just a plain presentation of the characteristics of patients with bronchiectasis who develop lung cancer. This is relatively not interesting. I have some additional remarks that should be addressed by the authors.

Response. We would like to apologize to the reviewer that we could not meet the standard of the reviewer. We fully agree with the reviewer that a study design comparing the risk of lung cancer in individuals with bronchiectasis versus those without bronchiectasis is very important. Regarding this subject, our analysis result is going to be published in a couple of days (Please see the attached acceptance mail from Annals of ATS). In our analyses, Individuals with bronchiectasis have a 22% increased risk of lung cancer compared to those without bronchiectasis even after adjusting for potential confounders.

     Meanwhile, we would like to say that evaluating potential risk factors for lung cancer development in bronchiectasis is also important since this study could provide meaningful data for health policy developers and physicians to make a preventive strategy and properly manage patients, respectively. From this perspective, again, we would like to suggest that this study has its own value to be published.

Comment 2. Major remarks. What were the criteria for diagnosis of lung cancer. Was the disease confirmed by a cytology/histology in every case? What criteria were followed to define the COPD and bronchiectasis? Is there a specific obligation by standards of care to diagnose those diseases?

R1-1. Lung cancer

Thank you for your comment. The diagnostic criteria of lung cancer are ICD-10 diagnosis code C33–C34 plus the specific insurance code for lung cancer (V193).

Since we do not have data on pathology, we could not confirm whether every lung cancer was confirmed by pathology (cytology/histology). However, it is likely that most cases might have been diagnosed based on pathologic results. Generally, the National Cancer Registry provides V193 code only when pathologic results (cytology or histology) are provided. However, in very few cases in which biopsy or cytology cannot be performed (DNR status or seriously ill status), the National Cancer Registry provides V193 code after a comprehensive review of medical records including medical charts and imaging information (CT, PET-CT, etc.). We added this information in the Discussion section (page10, 2nd paragraph)

“First, we used ICD-10 codes (C33–34) and a specific insurance code (V193 code) for the diagnosis of lung cancer. In Korea, once a person is diagnosed with lung cancer, he or she is registered with the National Cancer Registry and receives special insurance benefits. Thus, the validity of cancer diagnosis is strictly reviewed by the health insurance review and assessment service. Generally, the National Cancer Registry provides V193 code when pathologic results are provided. However, in very few cases in which biopsy or cytology cannot be performed (e.g., seriously ill status or Do-Not-Resuscitate status), the National Cancer Registry provides V193 code after a comprehensive review of medical records. Thus, there might be a chance of misclassification of lung cancer in patients who did not undergo pathological diagnostic approach.”

R2-2. Bronchiectasis and COPD

This study defined bronchiectasis and COPD by ICD-10 diagnosis codes. Therefore, the two diseases diagnosis do not require a specific obligation from physicians.  We have clarified this misclassification issue as a limitation in the Discussion section of the revised manuscript (page 10, 2nd paragraph).

“Second, we used ICD-10 codes for the diagnosis of bronchiectasis and comorbidities. Thus, there might be a misclassification, which is a major limitation of all-claim based studies”
